# A PPARG Splice Variant in Granulosa Cells Is Associated with Polycystic Ovary Syndrome

**DOI:** 10.3390/jcm11247285

**Published:** 2022-12-08

**Authors:** Chao-Yi Shi, Jing-Jing Xu, Cheng Li, Jia-Le Yu, Yan-Ting Wu, He-Feng Huang

**Affiliations:** 1Obstetrics and Gynecology Hospital, Institute of Reproduction and Development, Fudan University, Shanghai 200032, China; 2Ningbo Women and Children’s Hospital, Ningbo 315012, China; 3International Peace Maternity and Child Health Hospital, Shanghai Jiao Tong University School of Medicine, Shanghai 200030, China; 4Research Units of Embryo Original Diseases, Chinese Academy of Medical Sciences (No. 2019RU056), Shanghai 200030, China; 5Shanghai Key Laboratory of Embryo Original Diseases, Shanghai 200030, China

**Keywords:** PCOS, PPARG, splice variant, proliferation, migration, apoptosis, lipid metabolism

## Abstract

Background: We explored whether there are splice variants (SVs) of peroxisome proliferator-activated receptor-gamma (PPARG) in polycystic ovary syndrome (PCOS) patients and its relationship with clinical features and KGN cell functions. Methods: We performed a study involving 153 women with PCOS and 153 age-matched controls. One type of PPARG SV was detected by SMARTer RACE. The correlations between PPARG SV expression levels, clinical features, and KGN cell functions were analyzed. The effect of the PPARG SV on the expression of important genes in metabolism-related pathways was explored by PCR array. Results: The expression of the PPARG SV in PCOS patients was significantly higher than that in the controls. Clinical features were more significant in the PCOS group with the SV. Compared with overexpression of PPARG, the overexpression of the PPARG SV inhibited the proliferation, migration, and apoptosis of KGN cells in vitro. The genes related to the PPARG SV were mainly involved in lipid metabolism. Conclusion: While granulosa cells contribute greatly to the development of follicles, our results suggest that the identified PPARG SV may regulate cell proliferation, migration, and apoptosis in granulosa cells, which could partially explain the mechanisms of ovulation dysfunction in PCOS. Further investigation of the utility of this PPARG SV as a biomarker for PCOS is warranted.

## 1. Introduction

Polycystic ovary syndrome (PCOS) is a common reproductive and endocrine disorder characterized by ovulation dysfunction, hyperandrogenism, and polycystic ovarian morphology [1]. The prevalence of PCOS has a wide range of 2.2–26%, based on studies conducted in different countries with various methodologies [2,3,4]. PCOS is the main cause of female anovulatory infertility. Due to the heterogeneity of clinical manifestations, the pathogenesis of PCOS has not yet been fully elucidated.

Our understanding of alternative splicing has been revolutionized by high-throughput sequencing-based methods and their applications in transcriptomics. In the context of physiologically normal and disease states, networks of functionally coordinated and biologically important alternative splicing events are continuously being discovered in an ever-increasing diversity of cell types [5]. Peroxisome proliferator-activated receptor-gamma (PPARG) is an important regulator of follicular development, ovarian steroid hormone synthesis/metabolism, cell cycle regulation, and lipid metabolism [6,7]. This molecule has been investigated quite intensively in recent years. Polymorphisms and mutations in the PPARG gene have been reported to positively correlate with body mass index (BMI) in obese and diabetic patients [8,9]. In addition, 30 splice variants (SVs) of PPARG have been identified from tumor cells, and some SVs are associated with insulin resistance and metabolic diseases [10,11].

To date, the molecular mechanism of follicular development disorder in PCOS ovaries is still unclear. The complex transition from primordial follicles to mature follicles relies on granulosa cells (GCs). During this critical period, the growth of oocytes depends on two-way communication between oocytes and GCs [12]. GCs act as bridges to regulate the exchange of information between the inside and outside of each oocyte; in addition, proliferation/apoptosis itself predicts the fate of eggs in the same follicle. Each step of these developmental processes involves changes in the structure and function of the follicle, thus requiring precise and coordinated adjustments to the genes that play key roles in follicle selection and maturation or follicle-to-luteal transition. The dysfunction of GCs may be related to the abnormal development of PCOS follicles. The growth and death of early antral follicles are key determinants of the continued maturation of the dominant follicle/the atresia of the secondary follicle. Many early antral follicles and antral follicles with diameters of 2–10 mm are present in PCOS ovaries. In PCOS patients, there are very few signs of follicle production and ovulation. This feature suggests that the follicles of PCOS ovaries develop abnormally and that the growth of the dominant follicles is stagnant [13]. Studies have shown that there are significant differences in the cell death and proliferation rates of GC populations in PCOS patients [14]. However, whether GC apoptosis is involved in the pathogenesis of PCOS is still controversial. Song et al. found that high insulin induces GC apoptosis [15]. Some previous studies have shown that some genes that are abnormally expressed in women with PCOS can promote cell proliferation in KGN cells or GCs and inhibit apoptosis [16,17] Follicle apoptosis may be caused by a variety of potential negative factors and can occur at any stage of follicular development.

Here, we report the skipping of exon 5 as a PPARG pre-mRNA SV in ovarian GCs of PCOS patients. Considering the relationship between PPARG and metabolism [18,19,20,21], we hypothesized that the PPARG SV is related to PCOS and serves as another potential biomarker for PCOS. The purpose of this study was to examine the associations between PPARG and the PPARG SV in GCs and PCOS in a large clinical setting, clarify the epigenetic role of this PPARG SV in PCOS pathogenesis, and shed new light on the treatment of PCOS.

## 2. Materials and Methods

### 2.1. Patient Recruitment and Sample Collection

The study population included patients who visited the Reproductive Medicine Centre of the International Peace Maternal and Child Health Hospital (IPMCH) of Shanghai Jiao Tong University from August 2016 to August 2018. PCOS patients were diagnosed according to the revised 2003 Rotterdam Consensus criteria and women undergoing in vitro fertilization (IVF), due to tubal-factor or male-factor infertility, were recruited. The patients in the control group had regular menstrual cycles, biphasic basal body temperatures, and normal ovarian morphology as assessed by ultrasound. Both groups excluded patients with systemic diseases. Each group consisted of 153 patients, and all people voluntarily joined this study with informed consent. The protocol used for controlled ovarian hyperstimulation was either the long agonist protocol or the gonadotropin-releasing hormone (GnRH) antagonist protocol. After obtaining written consent and approval from the ethics committee of IPMCH (GKLW2016-28), human GCs were isolated from patients. Clinical characteristics, such as age, BMI, the levels of sex hormones and the numbers of follicles ≥ 14 mm on the human chorionic gonadotropin (hCG) peak day, were collected and analyzed.

### 2.2. RNA Extraction and SMARTer^®^ RACE

Total RNA was extracted from the cells using RNAiso reagent (code No. 9109, TaKaRa, Dalian, China) and reverse-transcribed by applying a PrimeScript RT Reagent Kit with gDNA eraser (Code No. RR047, TaKaRa, Dalian, China). To determine the SV, gene-specific primers were designed to amplify the full-length cDNA sequence of PPARG in GCs of clinical patients using a SMARTer^®^ RACE cDNA Amplification Kit (code No. 634858, TaKaRa Bio, Mountain View, CA, USA). Nested polymerase chain reaction (PCR) was used to increase the sensitivity and/or specificity of PCR. The results were verified by sequencing. The gene-specific primer sequences are shown in Appendix A.

### 2.3. Quantitative Reverse Transcription PCR (qRT–PCR)

RNA was extracted from patients’ GCs and reverse-transcribed into cDNA. Then, the expression of PPARG and its SV was immediately evaluated by PCR. qRT–PCR was carried out using SYBR Green Fast qPCR Mix (code No. RR420, TaKaRa, Dalian, China) under standard conditions (95 °C for 30 s and 40 cycles of 95 °C for 5 s and 60 °C for 34 s) on a QuantStudio 7 Flex instrument (Applied Biosystems, Foster City, CA, USA). The expression of the genes of interest was normalized to that of beta-actin and quantified according to the 2^−ΔΔCT^) method. The mRNA expression of PPARG and its SV was compared between the two groups using the Mann–Whitney U test and Student’s t test, respectively. The reactions were carried out in duplicate. The primer sequences are also given in Appendix A.

### 2.4. Cell Treatments

The human granulosa-like tumor cell line KGN was obtained from Riken Gene Bank (Tsukuba, Japan) [22]. The KGN cells were cultured in DMEM-F12 (code No. 11320033, Gibco, San Diego, CA, USA) supplemented with 10% Australian fetal bovine serum (FBS) (code No. 110099141, Gibco, San Diego, CA, USA), 100 U/mL penicillin G, 0.1 mg/mL streptomycin (code No. 15140163, Gibco, San Diego, CA, USA), and nonessential amino acids (code No. 11140076, Gibco, CA, USA). The cells were cultured in a 37 °C, 5% CO_2_ incubator.

### 2.5. RNA Interference and Overexpression of Candidate Genes

KGN cells with overexpression of PPARG and a PPARG deletion SV were constructed by lentiviral infection, and the control group consisted of KGN cells infected with a virus carrying an empty vector. Wild-type (WT) cells corresponded to the control group, rPPARG cells corresponded to the PPARG overexpression group, and shPPARG + rPPARG SV cells corresponded to the PPARG deletion SV overexpression group. To generate rRNA and shRNA of PPARG, KGN cells were transfected with a GV358 rRNA vector or GV248 shRNA vector with phU6, pMCS Ubiquitin-IRES, and puromycin. After transfection, cells were cultured in medium with 20% FBS. The coding sequence of the rPPARG SV was cloned into the LV6 vector with psi-RRE-EF1a-puromycin-WPRE and then co-transfected with shPPARG lentiviruses into KGN cells to produce lentiviruses overexpressing the PPARG SV and weakly expressing normal PPARG. All constructs were verified by sequencing. The vectors are shown in Appendix A.

### 2.6. Proliferation Assay

For the cell proliferation assay, cells were harvested, and cells on coverslips were prepared for bromodeoxyuridine (BrdU) labeling and immunostaining. Solutions of BrdU (10 mg/mL) were dissolved in phosphate-buffered saline (PBS). The slides were treated in BrdU for 2 h. After subsequent washing in TBS, DNA was denatured in HCl (2 mol/L, 15 min at 37 °C), pH-neutralized in sodium borate buffer (pH 8.5) and washed with PBS. The slides were incubated with a mouse anti-BrdU monoclonal antibody (code No. MA1-81890, Thermo Fisher, Waltham, MA, USA, 1:50 in TBS with 1% BSA) at 4 °C overnight in the dark and then co-incubated with secondary antibodies. The slides were counterstained with antifade mountant with DAPI (code No. P36970, Invitrogen, Waltham, MA, USA) for 10 min at room temperature in the dark. Fluorescence images were acquired using an inverted fluorescence microscope (Leica, Wetzlar, Germany).

Another proliferation assay was performed using Cell Counting Kit-8 (CCK-8) (Code No. E606335, Sangon Biotech, Shanghai, China). Cell suspensions (100 µL/well) and a certain number of cells (2000 cells/well) were inoculated into a 96-well plate. The cells were cultured in 5 wells, and enough cells were prepared for five straight days of measurements. Ten microliters of CCK-8 solution were added to each well of the plate. After 1–4 h of incubation, the absorbance at 450 nm was measured using a microplate reader (BioTek, Winooski, VT, USA). In addition, a colony formation assay was used to evaluate cell colony formation. A total of 2000 cells were incubated in 6-well plates until clones were visible to the human eye.

### 2.7. Wound Healing Assay

A total of 1 × 10^5^ KGN cells were plated onto 6-well plates until confluence. After 24 h, wounds were created by scratching the cell sheets with a sterile 1000 μL pipette tip. The KGN cells were washed in PBS. The culture medium was replaced with fresh DMEM/F-12 medium. Images of a specific position on the scratched areas were obtained with an inverted microscope (Leica, Wetzlar, Germany) every 6 h. The wound widths were measured.

### 2.8. Western Blot Analysis

Cells were homogenized in cell lysis buffer for Western blotting and immunoprecipitation (code No. P001, Beyotime, Shanghai, China). The homogenate was incubated on ice for 10 min and centrifuged at 12,000× *g* for 15 min. The protein concentration in the supernatant was quantified using a BCA Protein Assay Kit (code No. 23227, Thermo Fisher Scientific, USA). Samples (20 μg/lane) were separated on a 12% SDS–polyacrylamide gel. The separated samples were transferred to polyvinylidene fluoride (PVDF) membranes, incubated with primary antibodies (listed in Appendix A) at 4 °C overnight, incubated with horseradish peroxidase-conjugated immunoglobulin G for 1 h at room temperature, and visualized using enhanced chemiluminescence detection reagent (code No. WBKLS0500, Millipore Corporation, Billerica, MA, USA). The signals were captured, and the intensity of the bands was quantified using the Amersham Imager 600 image system (General Electric Company, Boston, MA, USA).

### 2.9. PCR Array

PCR array experiments were performed using a PCR Array Human PPAR Targets Kit (code No. 330231, Qiagen, Germantown, MD, USA). The gene expression values were calculated. Then, the fold change in the expression of each gene was calculated by dividing the gene expression value in the test sample by the gene expression value in the control sample. That is, the gene expression value of the control sample was normalized to 1. Next, the significantly differentially expressed genes were screened out: a gene was considered to be upregulated when the fold-change value was more than 2, but downregulated when the fold-change value was less than 0.5 (*p* < 0.05). The significantly differentially expressed genes were further investigated by bioinformatics analysis, and their influences on lipid metabolism were analyzed.

### 2.10. Statistical Analysis

The results of multiple experiments are presented as the mean ± SEM or SD. Statistical analyses were performed using GraphPad 6.0 statistical software (GraphPad, San Diego, CA, USA) and ImageJ (National Institutes of Health, Bethesda, MD, USA). N is the number of tissue preparations, cells, or separate experiments, as indicated in the figure legends. The *p* values were calculated using unpaired two-tailed *t*-tests or one-way analysis of variance (ANOVA), followed by the Bonferroni post-hoc test. A *p* value of <0.05 was considered to indicate a statistically significant result.

## 3. Results

### 3.1. Increased mRNA Expression of the PPARG SV in PCOS Women

From August 2016 to August 2018, 153 women with PCOS and 153 controls were included in this study. A SMARTer^®^ RACE 5′/3′ Kit and nested qRT–PCR were used to amplify PPARG cDNA from GCs of Han Chinese women undergoing IVF. We identified one alternative SV expressed at aberrantly high levels in women with PCOS. Through statistical analysis, we found that the BMI, the luteinizing hormone (LH)/follicle-stimulating hormone (FSH) ratio, the basal level of total testosterone (TT), and the number of ovulation-promoting > 14 mm follicles were significantly different between the PCOS groups and the control group, and the difference was more significant in the PCOS group with the SV than in the PCOS group without the SV (Table 1). The SV was determined to be an exon 5 deletion (PPARG SV). This was demonstrated by agarose gel electrophoresis (Figure 1A) and PCR product sequencing (Figure 1B). The schematic diagram of the splice variant and locations of primers for the identification of the presence or absence of exon 5 are shown in Figure 1C. The PPARG SV was expressed in 76.47% of women with PCOS, while it was detected in 50.98% of the controls (Figure 1D). As shown in Figure 1E, notably, the relative expression of PPARG SV was extremely high in women with PCOS, nearly quadruple the average in the control group. Taken together, these results suggest that the PPARG SV is associated with PCOS.

### 3.2. Overexpression of the PPARG SV Inhibits KGN Cell Proliferation

Then, we used the human granulosa-like tumor cell line KGN for subsequent in vitro experiments. KGN cells were transfected with either a vector with rPPARG or a vector with shPPARG + rPPARG SV, and empty vector-transfected KGN cells served as the controls (WT group). Then, we determined KGN cell proliferation by BrdU staining, CCK-8, and colony formation assay and found that overexpression of the PPARG SV significantly decreased cell proliferation (*p* < 0.05), compared to that in the WT group (Figure 2). Overexpression of PPARG had the opposite result.

### 3.3. Overexpression of the PPARG SV Inhibits KGN Cell Migration

Then, we measured the direct effect of the PPARG SV on KGN cell migration progression. For this, we made scratches on the cell culture plate when the cells reached confluency and observed the single-cell cloning progress. As shown in Figure 3A,B, overexpression of the PPARG SV decreased the cell migration progress.

### 3.4. Overexpression of the PPARG SV Inhibits KGN Cell Apoptosis

Next, we evaluated the effect of the PPARG SV on the caspase expression in KGN cells using Western blot analysis. Interestingly, as shown in Figure 4, the overexpression of the PPARG SV significantly downregulated the protein expression of caspase-3, caspase-8, and caspase-9, compared to that in the WT group. We performed an analysis of apoptosis by cytometry using TUNEL assay. The results showed that, compared with the WT group, the rPPARG group showed a significant increase in apoptosis (84.7% vs. 54.7%). However, although the number of apoptotic cells in the shPPARG + rPPARG SV group was lower than that in the WT group, the difference was not significant (47.8% vs. 54.7%). We have added them into Appendix A.

### 3.5. Screening of Significantly Differentially Expressed Genes in the PCR Array

According to the SV RNA sequence, we created a protein structure map (Figure 5). With a PPAR Signaling Pathway Plus PCR Array, the related gene expression profiles of PPARG and the PPARG SV were compared. Genes with significant differential expression were analyzed. The results showed that the overexpression of the PPARG deletion SV increased the expression of FABP3 and reduced the expression of APOC3, HELZ2, EP300, SLC27 family members, PCK2, CYP7A1, MMP9, β2-microglobulin, and ACSL5. Gene ontology (GO) and Kyoto Encyclopedia of Genes and Genomes (KEGG) enrichment analysis showed that these genes were mainly involved in the PPAR signaling pathway, adipocytokine signaling pathway, and glucagon signaling pathway. Analysis of the supplied data yielded a list of expression changes in the samples. The detailed results for the up- and downregulated genes associated with the PPAR signaling pathway are summarized in Table 2, Table 3, Table 4 and Table 5 and Figure 6.

## 4. Discussion

Using collected clinical data and samples, we amplified the full-length sequence of PPARG through RACE technology and discovered a deletion SV of PPARG. According to the NCBI database, it matches Homo sapiens peroxisome proliferator activated receptor gamma (PPARG), transcript variant 5, mRNA (NCBI reference sequence: NM_001330615.4). Then, we designed specific primers for the exon 5 deletion SV sequence and found that the PPARG exon 5 deletion SV was present in both the PCOS group and the control group. The PCOS group exhibited a higher relative expression of the SV than the control group. Subsequently, by constructing an in vitro cell model, we found that the PPARG exon 5 deletion SV has important effects on the proliferation, migration, apoptosis, and fat metabolism of GCs.

As a transcription factor, PPARG governs distinct biological processes through the modulation of target gene expression, mainly via ligand-dependent activity [23]. Ligand-dependent and ligand-independent activation mechanisms, large quantities of co-factors, and posttranslational and epigenetic modifications lead to the activation of distinct signal transduction pathways, resulting in cell- or tissue-specific responses [24]. Differential usage of promoters and alternative splicing generate transcripts with different mRNA stability and translational regulation, as well as with unique spatial and temporal expression.

As a result of maternal development, GCs can be used as a simple and representative system for studying oocyte development, follicular development, and related regulatory mechanisms. Our research group previously discovered androgen receptor SVs in PCOS patients, revealing the role of epigenetic modification in the pathogenesis of PCOS [25]. In the current study, we observed higher levels of the PPARG SV in the GCs of PCOS patients than in those of the control subjects. The PPARG SV corresponded to higher androgen levels, LH/FSH ratios, and BMI values. Consistent with evidence linking PPARG with PCOS [26,27,28,29], our findings indicate that the PPARG exon 5 deletion SV is a risk factor for PCOS.

Based on our findings, we hypothesize that the epigenetic mechanism of PPARG splicing abnormalities is involved in the pathophysiological process of PCOS. The specific mechanism is as follows: the PPARG mRNA precursor is alternatively spliced in ovarian GCs, and the cis-response element and trans-response element induce the differential expression of different PPARG SVs. Any event that hampers PPARG activity causes a loss of the differentiated phenotype. In fact, PPARG plays pivotal roles in adipocyte and epithelial cell differentiation, as well as in the regulation of energy metabolism [29]. As a transcription factor, PPARG is composed of a DNA-binding domain, a ligand-binding domain, a hinge region, and an N-terminal domain. PPARG generally binds with the DNA promoter region of the target gene to promote transcription. The ligand-binding domain is encoded by exons 5 and 6. The function of the protein is largely determined by its three-dimensional structure. The predicted three-dimensional structures of PPARG and the PPARG exon 5 deletion SV are obviously different. The lack of exon 5 in the PPARG SV influences the ligand-binding domain. Deleted SVs encode proteins different from WT PPARG, with the differences including fewer amino acids and altered three-dimensional structures. These changes may affect the function and activity of the PPARG protein, and they ultimately affect the transcription of downstream target genes, the proliferation of GCs, apoptosis, and gap junction communication, leading to PCOS-related follicular development arrest. Whether PPARG or its exon 5 deletion SV plays other roles or even becomes another protein completely to play a role in a pathway other than the PPARG pathway, which needs to be researched further.

PPARG is the main target of thiazolidinedione (TZD)-based insulin-sensitizing therapies. Notably, the PPARG exon 5 deletion SV positively correlated with BMI. Therefore, high PPARG exon 5 deletion SV levels in the GCs of obese individuals may account for the reduced transcription of metabolic genes and the impaired lipid metabolism in these individuals, which are strictly correlated with insulin resistance [30,31]. However, such results often only start an investigation into the underlying mechanisms at work. In order to assist in further analysis, the latest bioinformatics tools should be used to analyze the data and suggest regulatory mechanisms and future experiments.

Our study reveals that the PPARG exon 5 deletion SV in GCs inhibits cell proliferation, migration, and apoptosis and may ultimately affect the development of follicles, which provides a basis for revealing the relevant mechanisms of follicular maturation disorders in PCOS ovaries. Although our study was limited to eastern Han women in China, the results suggest the epigenetic role of the PPARG exon 5 deletion SV in PCOS pathogenesis and provide new insights for the treatment of PCOS. Our findings suggest that epigenetics should be explored further, and it promotes progress in the new field of PCOS pathogenesis research. However, similar experiments need to be performed in a number of mouse models. Besides, as there are eight variants of the PPARG gene that lack exon 5, further research should focus on each specific splice variant. Considerably more work will hopefully be performed in this area.

## 5. Conclusions

In summary, the present study demonstrates, for the first time, the contribution of a PPARG SV to PCOS. These findings establish alternative splicing as a paradigm of PPARG regulation, opening new avenues for research on the roles of PPARG in physiological processes and/or pathological conditions. Further study is warranted to elucidate the molecular mechanisms underlying this association.

## Figures and Tables

**Figure 1 jcm-11-07285-f001:**
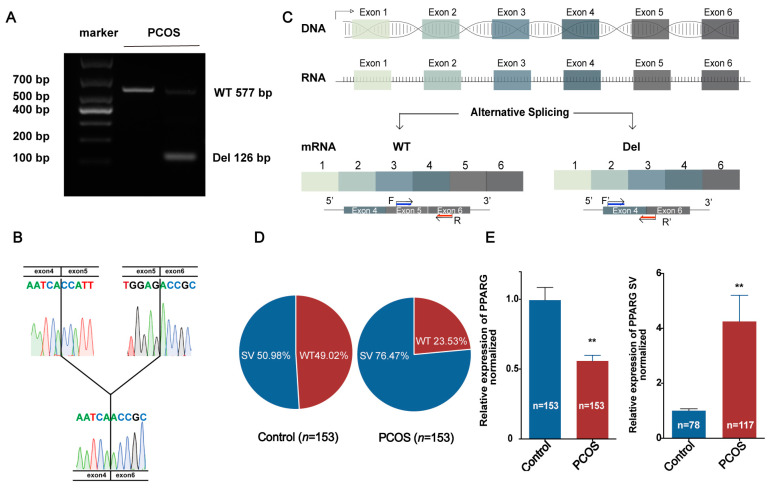
Identification and genomic origin of the PPARG deletion splice variant (del PPARG SV) in the granulosa cells (GCs) of women in the PCOS and control groups. (**A**–**C**) Representative PCR results (**A**) and corresponding sequencing results for the PPARG SV in human GCs (**B**,**C**), locations of primers for identification of the presence or absence of exon 5 (**C**). (**D**) Percent distribution of the PPARG SV in control subjects (*n* = 153) and patients with PCOS (*n* = 153). (**E**) qRT–PCR was used to evaluate the relative expression levels of PPARG and its SV in GCs of the control group and the PCOS group with PPARG SV expression. The data are presented as the mean ± SEM, ** *p* < 0.01. qRT–PCR: quantitative reverse transcription polymerase chain reaction; GC: granulosa cell.

**Figure 2 jcm-11-07285-f002:**
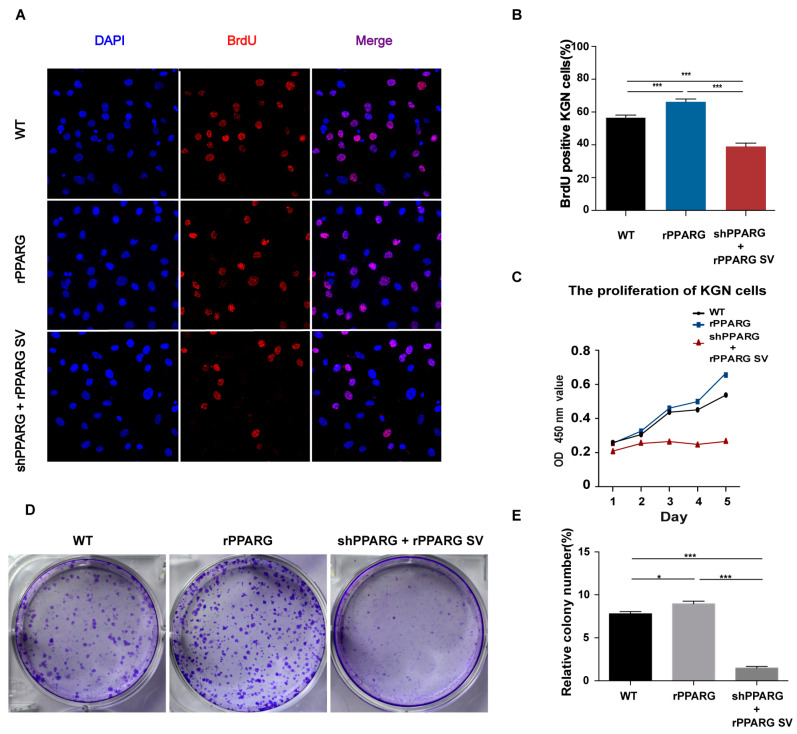
Measurements of cell proliferation in KGN cells. (**A**,**B**) BrdU staining was performed to measure KGN cells, and the BrdU-positive cells were calculated. Scale bar: 100 μm. (**C**) A CCK-8 assay showed that PPARG downregulation and overexpression of its SV significantly impeded the proliferation of KGN cell lines. (**D**,**E**) Colony formation assays showed significant differences among the three cell groups. The data are presented as the mean ± SD. * *p* < 0.05, *** *p* < 0.001.

**Figure 3 jcm-11-07285-f003:**
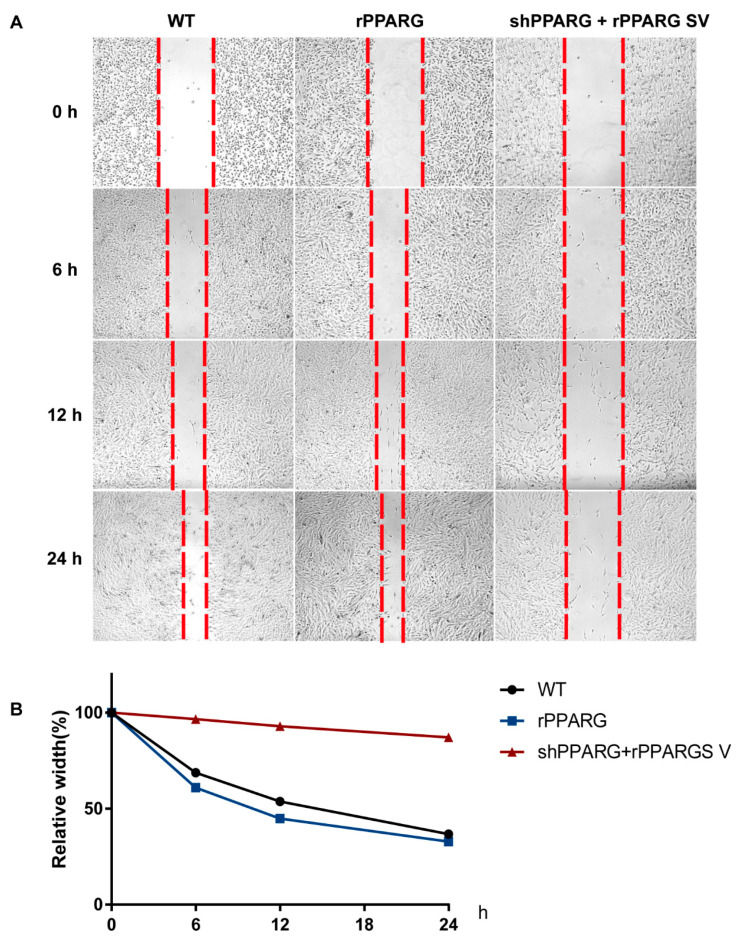
(**A**,**B**) Cell migration was measured using a wound healing assay, and the relative wound widths of the wounded cells were measured at 0, 6, 12, and 24 h after wound injury. The data are presented as the mean ± SD.

**Figure 4 jcm-11-07285-f004:**
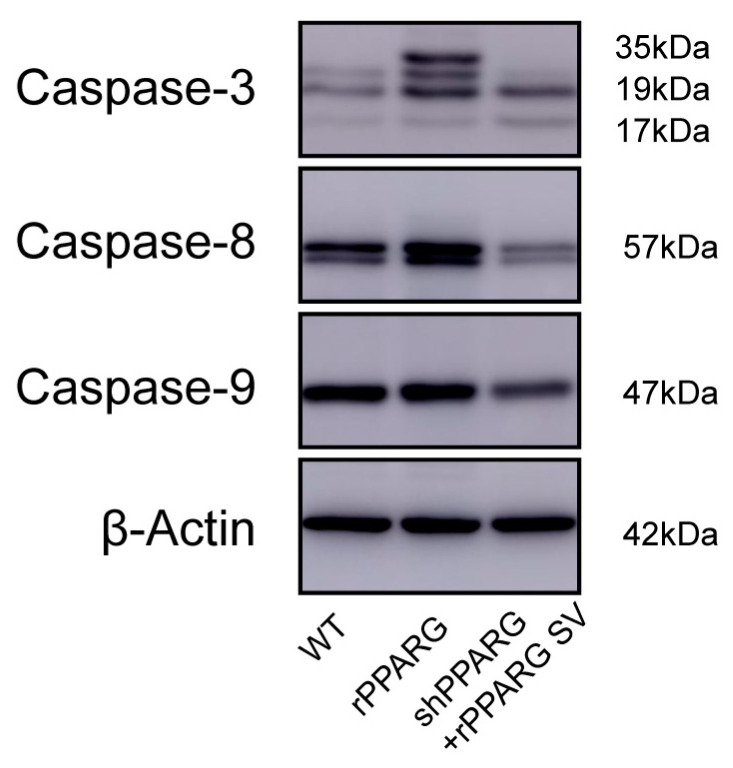
Measurements of cell apoptosis in KGN cells. The protein expression of caspase-3, caspase-8, and caspase-9 in KGN cells in each group was detected by Western blot analysis. The figure shows caspase-3 (17, 19, and 35 kDa), caspase-8 (57 kDa), caspase-9 (47 kDa), and β-actin (42 kDa) immunoreactive bands.

**Figure 5 jcm-11-07285-f005:**
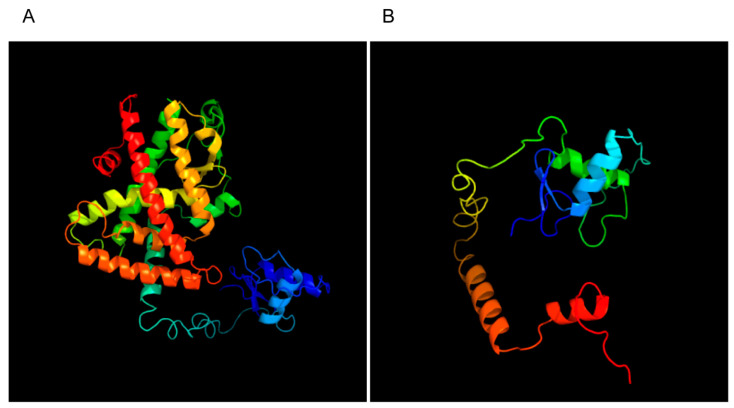
PPARG SV protein structure prediction. (**A**) Normal PPARG protein structure. (**B**) PPARG SV protein structure.

**Figure 6 jcm-11-07285-f006:**
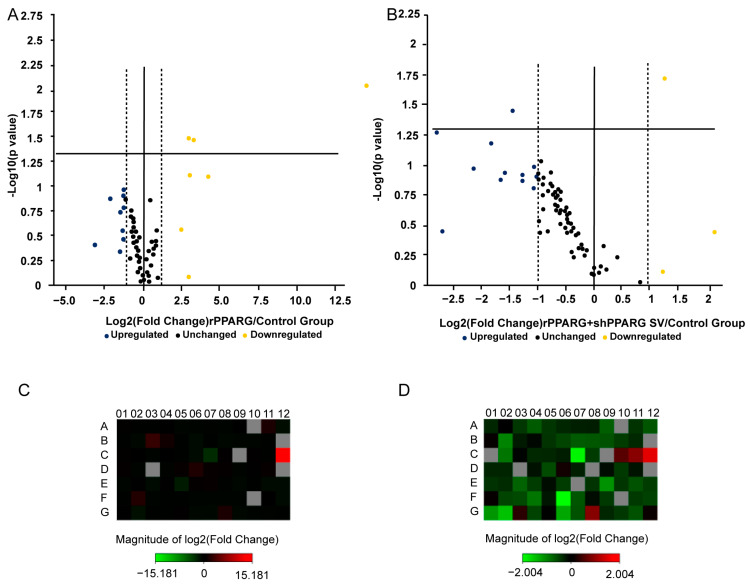
Results of PCR array. (**A**) Genes upregulated and downregulated in the rPPARG group vs. the control group (WT) (scatter plot). (**B**) Genes upregulated and downregulated in the shPPARG +rPPARG SV group vs. the control group (WT) (scatter plot). (**C**) PCR array heatmap of the rPPARG group vs. the control group (WT). (**D**) PCR array heatmap of the shPPARG +rPPARG SV group vs. the control group (WT).

**Table 1 jcm-11-07285-t001:** Clinical characteristics of women with PCOS and control subjects.

Clinical Parameter	Control		PCOS	
	WT *	SV #	WT	SV
*n*	75	78	36	117
Age (year)	30.48 ± 0.37	30.59 ± 0.37	30.69 ± 0.51	30.41 ± 0.28
BMI (kg/m^2^)	22.35 ± 0.36	22.38 ± 0.37	24.94 ± 0.67 **	24.99 ± 0.28 ##
Duration of infertility (year)	3.57 ± 0.22	3.78 ± 0.25	3.77 ± 0.40	3.66 ± 0.20
Day 3 serum LH/FSH	0.79 ± 0.12	0.73 ± 0.08	1.26 ± 0.14 *	1.56 ± 0.08 ##
Day 3 serum TT (nmol/L)	1.62 ± 0.07	1.61 ± 0.08	2.19 ± 0.11 **	2.44 ± 0.10 ##
Day 3 serum E2 (pmol/L)	185.60 ± 11.58	182.10 ± 11.74	195.50 ± 17.43	168.60 ± 7.71
No. of >14 mm follicle	9.18 ± 0.61	8.98 ± 0.61	9.66 ± 1.04	13.08 ± 0.69 ##

Data were shown as the mean ± SEM. * *p* < 0.05, ** *p* < 0.01, compared with the control WT group. # *p* < 0.05, ## *p* < 0.01, compared with the control SV group.

**Table 2 jcm-11-07285-t002:** Genes Over-expressed in Group 1 (rPPARG) vs. Control Group (WT).

Position	Gene Symbol	Fold Regulation	RT2 Catalog
B03	APOE	7.39	PPH01366D
F02	PPARG	3.98	PPH01366D
D06	GK	3.35	PPH01092A
A11	ANGPTL4	3.22	PPH02234F
G08	SORBS1	3.14	PPH11098B
B04	CD36	2.28	PPH01356A

**Table 3 jcm-11-07285-t003:** Genes Under-expressed in Group 1 (rPPARG) vs. Control Group (WT).

Position	Gene Symbol	Fold Regulation	RT2 Catalog
C07	EP300	−6.84	PPH00319A
E05	NCOA6	−3.62	PPH05909A
G02	SLC27A1	−2.42	PPH17902A
A12	APOA1	−2.37	PPH02633B
E11	PLTP	−2.16	PPH01426A
C06	ELN	−2.11	PPH06895F
F12	SIRT1	−2.11	PPH02188A
B06	CLU	−2.09	PPH00243F

**Table 4 jcm-11-07285-t004:** Genes Over-expressed in Group 2 (shPPARG + rPPARG SV) vs. Control Group (WT).

Position	Gene Symbol	Fold Regulation	RT2 Catalog
C12	FABP4	4.45	PPH02382F
C11	FABP3	2.42	PPH02460C
G08	SORBS1	2.37	PPH11098B

**Table 5 jcm-11-07285-t005:** Genes Under-expressed in Group 2 (shPPARG + rPPARG SV) vs. Control Group (WT).

Position	Gene Symbol	Fold Regulation	RT2 Catalog
F06	HELZ2	−6.98	PPH02218A
C07	EP300	−6.54	PPH00319A
G02	SLC27A1	−4.42	PPH17902A
G06	SLC27A6	−3.54	PPH09394A
G01	SLC22A5	−3.16	PPH08052A
E09	PCK2	−3.02	PPH02080B
B02	APOC3	−2.73	PPH01996A
C02	CYP7A1	−2.42	PPH01231A
F04	PPARGC1B	−2.41	PPH00030A
E03	MMP9	−2.11	PPH00152E
H02	B2M	−2.09	PPH01094E
A09	ACSL5	−2.03	PPH06335A

## Data Availability

The datasets analyzed during the current study are available from the corresponding author on reasonable request.

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
