# Peer review of "A PPARG Splice Variant in Granulosa Cells Is Associated with Polycystic Ovary Syndrome"

_jcm, 2022, doi:10.3390/jcm11247285_

Round 1

Reviewer 1 Report

According to the Genecards database (https://www.genecards.org/cgi-bin/carddisp.pl?gene=PPARG), there are 9 alternative splice variants of the PPARG gene, among which 8 variants are without exon 5. The authors of the manuscript are talking about one alternative spliced variant without exon 5. So which of the 8 splice variants was analyzed by the authors? What do the authors of the manuscript mean when they talk about the full-length mRNA of the PPARG gene? No spliced mRNA variant of the PPARG gene has all 13 exons. I strongly recommend drawing a diagram of alternative mRNA variants of the gene taken from the Genecards database (https://www.genecards.org/cgi-bin/carddisp.pl?gene=PPARG) and indicating on it the location of the primers used in PCR from the Table S1. In the light of the above, what were lentiviral constructs in the KGN cells for overexpression of PPARG and a PPARG deletion SV, and according to what sequences a protein structure map (Figure 5) was created?

Minor revisions: there are a lot of typos. For example, “hagnitude” should be substituted for “magnitude” in the Figure 6C, 6D.

Author Response

Dear editors and reviewers,

Thank you for reviewing our manuscript entitled “A PPARG splice variant in granulosa cells is associated with polycystic ovary syndrome " (jcm-1986925). We very much appreciate the comments provided by the editors and reviewers. We have addressed all of the comments with a point-by-point response provided below. For each point we have also indicated where in the manuscript we have made the modification. In the manuscript we have marked the revised segments.

The revised manuscript has been approved by all authors. In addition, the language has been proof-read by AMERICAN JOURNAL EXPERT to improve the readability of the text and the authors are entirely responsible for the scientific content of the paper.

We thank you for your time to review our response. If there are any further modifications or questions, please do not hesitate to contact me.

Yours sincerely on behalf of all authors,

He-Feng Huang

Obstetrics and Gynecology Hospital, Institute of Reproduction and Development, Fudan University, Shanghai, China

RESPONSE TO REVIEWER COMMENTS

Reviewer #1

Reviewer #1, Comment #1&2

According to the Genecards database (https://www.genecards.org/cgi-bin/carddisp.pl?gene=PPARG), there are 9 alternative splice variants of the PPARG gene, among which 8 variants are without exon 5. The authors of the manuscript are talking about one alternative spliced variant without exon 5. So which of the 8 splice variants was analyzed by the authors? What do the authors of the manuscript mean when they talk about the full-length mRNA of the PPARG gene?

Response: Thank you for your comments. The sequence of PPARG splice variant we detected is listed in the supplementary materials. Due to the limited number of splice variants in GeneCard, we compared the sequence in NCBI. We found that the SV we detected is NCBI Reference Sequence: NM_001330615.4. In our manuscript, when we talk about the full-length mRNA of the PPARG gene, it relates to the mRNA with the presence or absence of exon 5, and the latter is PPARG transcript variant 5 according to the sequence. To avoid misunderstanding, we have added this to Discussion section.

(Please see Page 12, Line 312-315,Discussion section; supplementary materials)

Reviewer #1, Comment #3

No spliced mRNA variant of the PPARG gene has all 13 exons. I strongly recommend drawing a diagram of alternative mRNA variants of the gene taken from the Genecards database (https://www.genecards.org/cgi-bin/carddisp.pl?gene=PPARG) and indicating on it the location of the primers used in PCR from the Table S1.

Response: Thank you for your suggestion. The location of primers for PPARG and its exon 5 deletion SV is listed in the attachment. The reverse primer for PPARG exon 5 deletion SV locates at the junction of exon 4 and exon 6. We have marked the location of primers in Figure 1C.

(Please see Page 6 , Line 232, Figure 1C )

Reviewer #1, Comment #4&5

In the light of the above, what were lentiviral constructs in the KGN cells for overexpression of PPARG and a PPARG deletion SV, and according to what sequences a protein structure map (Figure 5) was created?

Response: Sorry for the confusion. We have added the construction process of lentiviral in the KGN cells for overexpression of PPARG and a PPARG deletion SV to supplementary materials. To generate rRNA and shRNA of PPARG, KGN cells were transfected with a GV358 rRNA vector or GV248 shRNA vector with phU6, pMCS Ubiquitin-IRES and puromycin. After transfection, cells were cultured in medium with 20% FBS. The details of lentiviral constructs are listed in the attachment.

As a transcription factor, PPARG is composed of DNA binding domain, ligand binding domain, hinge region, and N-terminal domain. PPARG is generally combined with the DNA promoter region of the target gene to promote transcription. The ligand-binding domain was encoded by exons 5 and 6. The function of the protein is largely determined by its three-dimensional structure. From the perspective of the prediction of the three-dimensional structure, the two structures are obviously different. Since PPARG SV lacks exon 5, the ligand-binding domain would be influenced. Whether PPARG or SV plays other roles, or even becomes another protein completely to play a role in the pathway other than PPARG, it needs further research in the future.

The deletion mutation resulted in premature termination of protein translation, which was shortened from 475 amino acids to 248 amino acids. (NCBI Reference Sequence:  NP_001317544.2)

(Please see Page 13, Line348-360 Discussion section, supplementary materials /additional materials in the file)

Reviewer #1, Comment #6

Minor revisions: there are a lot of typos. For example, “hagnitude” should be substituted for “magnitude” in the Figure 6C, 6D.

Response: Thank you for your suggestion. We have modified the clerical errors in the full manuscript.

(Please see Page 12 , Line 305, Figure 6)

Reviewer 2 Report

This manuscript involves investigations into the functions of a PPARG splice variant in granulosa cells in polycystic ovary syndrome. The authors found the expression of the PPARG SV was abnormally higher in GCs of PCOS patients. Clinical characteristics also prove the PPARG SV is associated with PCOS. The authors further identified that overexpression of the PPARG SV inhibits KGN cell proliferation, migration and inhibits KGN cell apoptosis.

This manuscript is generally well written, and the experiments were well designed and performed; however, there are some flaws which need to be addressed.

1.      The authors showed that overexpression of the PPARG SV inhibits KGN migration. As a cancer cell line derived from GC tumor, KGN has characteristic of migration, but so far as I know, it isn’t PCOS GC phenotype. I am confused why the authors chosen this experiment method.

2.      Lines 278: Figure 6A should modified the abscissa (e.g., set segments) to make the scatter plot clearer.

Author Response

Dear editors and reviewers,

Thank you for reviewing our manuscript entitled “A PPARG splice variant in granulosa cells is associated with polycystic ovary syndrome " (jcm-1986925). We very much appreciate the comments provided by the editors and reviewers. We have addressed all of the comments with a point-by-point response provided below. For each point we have also indicated where in the manuscript we have made the modification. In the manuscript we have marked the revised segments.

The revised manuscript has been approved by all authors. In addition, the language has been proof-read by AMERICAN JOURNAL EXPERT to improve the readability of the text and the authors are entirely responsible for the scientific content of the paper.

We thank you for your time to review our response. If there are any further modifications or questions, please do not hesitate to contact me.

Yours sincerely on behalf of all authors,

He-Feng Huang

Obstetrics and Gynecology Hospital, Institute of Reproduction and Development, Fudan University, Shanghai, China

Reviewer #2

This manuscript involves investigations into the functions of a PPARG splice variant in granulosa cells in polycystic ovary syndrome. The authors found the expression of the PPARG SV was abnormally higher in GCs of PCOS patients. Clinical characteristics also prove the PPARG SV is associated with PCOS. The authors further identified that overexpression of the PPARG SV inhibits KGN cell proliferation, migration and inhibits KGN cell apoptosis.

This manuscript is generally well written, and the experiments were well designed and performed; however, there are some flaws which need to be addressed.

Reviewer #2, Comment #1

The authors showed that overexpression of the PPARG SV inhibits KGN migration. As a cancer cell line derived from GC tumor, KGN has characteristic of migration, but so far as I know, it isn’t PCOS GC phenotype. I am confused why the authors chosen this experiment method.

Response:

Thank you for the suggestions about the manuscript. We have substantially revised our manuscript as your suggestions. Alterations in granulosa cells in the ovary may be an intrinsic mechanism of PCOS. Granulosa cells are closely related to oocyte development. During reproductive life, human ovarian tissue goes through a continuous and extensive remodeling process for follicular growth, ovulation and atresia. The abnormalities of endocrine and intra-ovarian paracrine interactions may change the microenvironment for oocyte development during the folliculogenesis process and reduce the developmental competence of oocytes in PCOS patients who are suffering from anovulatory infertility and pregnancy loss. In this microenvironment, the cross talk between an oocyte and the surrounding cumulus cells is critical for achieving oocyte competence. The maturation of follicles involves the transformation of the surrounding granulosa cells from monolayer to multilayer, which may be related to the migration of granulosa cells. We also briefly introduced the relationship between granulosa cells and follicle maturation in the introduction. There are many previous studies on the cellular function of granulosa cells in PCOS, and these studies also studied the migration of granulosa cells (PMID: 35938797, 29432996, 32220601).

References:

Xiang YG,Yu G, Song YX , et al.The Upregulation of HAS2-AS1 Relates to the Granulosa Cell Dysfunction by Repressing TGF-β Signaling and Upregulating HAS2. [Journal Article].Mol Cell Biol. 2022 Sep 15;42(9):e0010722.

Han QF, Zhang WK, Meng JL , Ma L , Li AH. LncRNA-LET inhibits cell viability, migration and EMT while induces apoptosis by up-regulation of TIMP2 in human granulosa-like tumor cell line KGN. [Journal Article]. Biomed Pharmacother. 2018 Apr;100:250-256. 

Jia CW , Wang SY ,Yin CH , et al. Loss of hsa_circ_0118530 inhibits human granulosa-like tumor cell line KGN cell injury by sponging miR-136 - ScienceDirect [Journal Article].. Gene, 744.

Reviewer #2, Comment #2

Lines 278: Figure 6A should modified the abscissa (e.g., set segments) to make the scatter plot clearer.

Response: Thank you for the suggestion. We have modified Figure 6 according to your advice.
(Please see Page 12 , Line 305 , Figure 6)
